# Towards Efficient and Scalable Training of Differentially Private Deep Learning

**Sebastian Rodriguez Beltran** [1]   **Marlon Tobaben** [1]   **Niki Loppi** [2]   **Antti Honkela** [1]

## Abstract

Differentially private stochastic gradient descent (DP-SGD) is the standard algorithm for training machine learning models under differential privacy (DP). The major drawback of DP-SGD is the drop in utility which prior work has comprehensively studied. However, in practice another major drawback that hinders the large-scale deployment is the significantly higher computational cost. We conduct a comprehensive empirical study to quantify the computational cost of training deep learning models under DP and benchmark methods that aim at reducing the cost. Among these are more efficient implementations of DP-SGD and training with lower precision. Finally, we study the scaling behaviour using up to 80 GPUs.

## 1. Introduction

Training data of machine learning (ML) models can be extracted (Balle et al., 2022; Carlini et al., 2021). Differential Privacy (DP) (Dwork et al., 2006) is the gold standard for formalizing the privacy leakage of training data in ML and mitigating the risk of privacy attacks on the training data. DP is deployed in many applications involving sensitive data (Abowd, 2018; Cormode et al., 2018).

The established algorithm for integrating DP into the training pipeline of deep learning models is DP stochastic gradient descent (DP-SGD) (Rajkumar & Agarwal, 2012; Song et al., 2013; Abadi et al., 2016), which is the DP adaptation of SGD (see also Algorithm 1). DP-SGD has two major drawbacks in comparison to SGD:

- **Higher computational cost:** DP-SGD requires more memory and is computationally more expensive due to

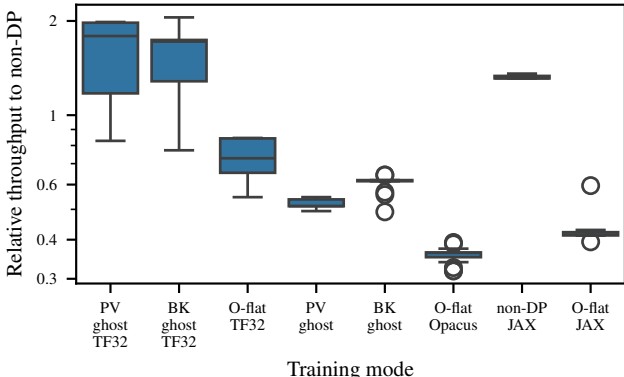

*Figure 1.* Relative throughput to the respective non private baseline. For each optimization method and each model size, we divide its throughput with the non-private counterpart. Throughput is the number of processed instances per second. The optimization mode includes clipping optimizations, lower precision, and frameworks like JAX.

the per-sample clipping.
- **Loss in utility:** The utility in comparison to non-DP training drops but this can be mitigated to certain extend by using larger batch sizes (Räisä et al., 2024) and training longer (Ponomareva et al., 2023) which further increases the computational cost.

**List of contributions** In this work we perform an extensive empirical study on the computational efficiency of DP-SGD. We will focus on fine-tuning a wide range of large computer vision classification models but our findings can be applied any other large models that are trained or fine-tuned with DP-SGD. Our main contributions are the following:

1. We find that non-optimized training with DP-SGD costs up to three and eight times more non-private training for ViT and ResNets, respectively (See Section 4).
2. We identify the reasons that lead to the higher computational cost of DP-SGD using profiling (See Section 4.3).
3. We benchmark different strategies that can reduce this cost drastically up to a level that matches the non-optimized non-private training:

   - More efficient implementations of DP-SGD (See Section 5.1).

[1]Department of Computer Science, University of Helsinki, Helsinki, Finland [2]NVIDIA, Helsinki, Finland. Correspondence to: Sebastian Rodriguez Beltran <sebastian.rodriguez@helsinki.fi>, Marlon Tobaben <marlon.tobaben@helsinki.fi>, Niki Loppi <nloppi@nvidia.com>, Antti Honkela .

Accepted to the Workshop on Advancing Neural Network Training at International Conference on Machine Learning (WANT@ICML 2024).

- Use of JAX instead of PyTorch (See Section 5.2).
- Lower Precision with TF-32 (See Section 5.3).

4. We scale up the training to 80 GPUs and find that DP-SGD scales better than non-private training (See Section 6).

## 2. Background

This section will explain the main DP-SGD algorithm and optimizations to alleviate its computational cost.

### 2.1. DP-SGD Algorithm

Algorithm 1 is the original DP-SGD algorithm, with virtual batching, as proposed by Abadi et al. (2016).

---
**Algorithm 1** Virtual Batching DP-SGD
---
**Input:** Training data points $\{x_1, \ldots, x_N\}$, loss function $\mathcal{L}(\theta) = \frac{1}{N} \sum_i \mathcal{L}(\theta, x_i)$
**Parameters:** Parameters: learning rate $\eta_t$, noise scale $\sigma$, gradient norm bound $C$, number of steps $T$, logical batch size $L$, physical batch size $p$.
**for** $t \in [T]$ **do**
    B ← Sample batch by Poisson sampling $L/N$.
    P ← Divide the logical batch B into physical batches of size $p$.
    $\theta_{acc} \leftarrow 0$
    **for** $s \in [P]$ **do**
        **Compute gradient**
        For each $i \in s$ compute $\mathbf{g}_t(x_i) \leftarrow \nabla_{\theta_t} \mathcal{L}(\theta_t, x_i)$
        **Clip gradient**
        $\overline{\mathbf{g}}_t(x_i) \leftarrow \mathbf{g}_t / \max(1, \frac{\|\mathbf{g}_t(x_i)\|_2}{C})$
        **Accumulate gradient**
        $\theta_{acc} \leftarrow \theta_{acc} + \sum_i \overline{\mathbf{g}}_t(x_i)$
    **end for**
    **Add noise**
    $\widetilde{\mathbf{g}}_t \leftarrow \frac{1}{|L|}(\theta_{acc} + \mathcal{N}(0, \sigma^2 C^2 \mathbf{I}))$
    **Step**
    $\theta_{t+1} \leftarrow \theta_t - \eta_t \widetilde{\mathbf{g}}_t$
**end for**
**Return** Learned parameters $\theta_T$ and then the privacy cost is computed.

---

**Virtual Batching** distinguishes between logical and physical batches. Logical batches are divided into multiple physical batches to allow taking optimizer steps with many samples without running out of memory. In our experiments, we typically sample logical batch sizes of size $L = 25000$ while the memory only fits $< 300$ samples at a time. Implementing DP-SGD with virtual batching Algorithm 1 does not modify the privacy accounting. The amount of added noise is the same and does not affect the model utility (Ponomareva et al., 2023).

**Opacus** Our baseline uses the PyTorch (Ansel et al., 2024) library Opacus (Yousefpour et al., 2021). Opacus is the most mature DP-SGD framework out of all considered implementations. It supports nearly all neural layers that are compatible with DP training. Opacus implements per-sample clipping without any additional optimizations. Opacus implements virtual batching algorithm in their `BatchMemoryManager`. The privacy engine of Opacus will sample the logical batches, as in the original DP-SGD algorithm, and then divide them into physical batches. The other implementations considered in our experiments do not support virtual batching out-of-the-box.

**Poisson subsampling** Interestingly, Bu et al. (2022) and Bu et al. (2023) never mention Poisson subsampling in their works of Mix Ghost Clipping and Book Keeping. Even more, Bu et al. (2022) states that it has a speed-up of $\times 1.7$ times against other algorithms with a fixed batch size, which would affect the privacy accountant method. The same happens in practice for JAX implementations (De et al., 2022), were the sampling is done by shuffling the dataset and using each sample once per epoch. While it is easier to implement in practice, it does not use the correct Poisson subsampling for the numerical accounting methods.

To make a fair comparison between all methods, we implement the Poisson subsampling, the same way Opacus does it, for all frameworks and a new custom Batch Memory Manager to flag when it is time to take a step. This way, all the experiments are seeded and will have the same logical and physical batch sizes.

### 2.2. DP-SGD Optimizations

We benchmark five types of clipping methods. Table A1 shows which clipping optimizations we are benchmark against the library or framework that implements it.

**JAX** We compare all implementations with a native JAX (Bradbury et al., 2018) implementation that clips the per-sample gradients with Optax (DeepMind et al., 2020) without utilizing any further optimization.

**Ghost clipping** computes the gradient norm loss after the backpropagation optimization and then reweights the loss to update the clipped gradients. Its main contribution is memory saving at the cost of adding another backward pass (Li et al., 2022).

**Mixed Ghost clipping** proposed by Bu et al. (2022) builds on-top of Ghost clipping. It implements the ghost clipping technique for convolutional layers. Its main contribution is that the algorithm will decide when to clip the gradients using per-example or ghost. This difference matters because the ghost clipping is less efficient when the layer's input dimensions are too big. E.g., for ResNets, each clipping method will be applied for half of the layers. The first layers

will be clipped using the the per-example and then ghost clipping in the bottom layers. As the model goes deeper, the feature size decreases, and the number of channels increases, prioritizing ghost clipping (Bu et al., 2022).

**Book keeping** by Bu et al. (2023) uses all the previous techniques but requires only one backpropagation pass without explicitly calculating the per-example gradients. It avoids the second pass that ghost clipping does by reusing the intermediate results of the output gradients to calculate the sum of the clipped gradients and the clipping factor. Book keeping can also be implemented together with the mix optimization. It also implements another technique called MixOpt, which does the same evaluation as the mix ghost clipping, but also determines whether doing a second backward pass is more efficient.

**Implementations of optimized DP-SGD** PrivateVision (Bu et al., 2022), and FastDP (Bu et al., 2023) are PyTorch-based implementations. While both PrivateVision and FastDP implement ghost clipping and its variants, all FastDP implementations use the book keeping method.

## 3. Experiment Overview

In our experiments we compare the throughput, defined as how many samples can be processed per second during training.

**Datasets** We benchmark with the dataset CIFAR100 (Krizhevsky & Hinton, 2009). It contains 60,000 data points, from 100 classes. It is a commonly used dataset for testing computer vision models and their private counterparts (Abadi et al., 2016; De et al., 2022; Yousefpour et al., 2021).

**Models** We benchmark two families of models. The Vision Transformer (ViT) (Dosovitskiy et al., 2021) and BiT-ResNet (Kolesnikov et al., 2020) (SeeTable 1). All models are pre-trained on the same public dataset, ImageNet-21k. However, the ViT models are additionally fine-tuned on ImageNet-1k (Steiner et al., 2022; Wightman, 2019). For our experiments, we are fine-tuning the feature encoder on CIFAR100. CIFAR-100 images are resized to 224x224.

**Parameterization** While parameter-efficient fine-tuning of some parts of the model, e.g. using adapters like FiLM Perez et al. (2018) or LoRA (Hu et al., 2022) has been shown to be effective under DP (Tobaben et al., 2023; Yu et al., 2022), our work focuses on the computational efficiency of DP-SGD and thus we consider on the worst-case scenario which is fine-tuning all parameters of the model. Furthermore, any training from scratch requires training all parameters.

**Hyperparameters** All experiments use the optimal hyperparameters found previously by Tobaben et al. (2023) with a privacy budget of $\epsilon = 8$. We train for two epochs, and according to the sampling rate and batch sizes, there will be

*Table 1.* Number of parameters, in millions, for each family architecture and size of the model.

| MODEL | | # OF PARAMETERS |
|---|---|---|
| | TINY | 5.7 M |
| | SMALL | 22.1 M |
| ViT | BASE | 86.6 M |
| | LARGE | 304.3 M |
| | HUGE | 630.8 M |
| | $50 \times 1$ | 23.7 M |
| | $101 \times 1$ | 42.7 M |
| BiT-ResNet | $50 \times 3$ | 211.8 M |
| | $101 \times 3$ | 382.4 M |
| | $152 \times 4$ | 929.2 M |

four optimizer steps in total. Using only two epochs allows us to test the experiments quickly and evaluate them over multiple runs. We do not focus on finding the best possible utility, which requires training for many more epochs. In Table A4 we show the mean accuracy after training for just two epochs. For optimal utility, it must be trained for a higher number of epochs.

**Environment specifications** We use two GPU architectures, NVIDIA V100 and NVIDIA A100, with 32 and 40 GB of VRAM respectively. Both use identical Python environments, duplicated in both to make a fair comparison. For the multinode experiments, each node has four GPU units. All experiments use 16 CPU workers, except the distributed DP, which breaks with more than one worker.

**Source code** We will make the source code for replicating the experiments available at `https://github.com/DPBayes/Towards-Efficient-Scalable-Training-DP-DL`.

## 4. What is the cost of DP in Deep Learning?

In this section we will quantify the computational cost of deploying DP training as a whole. We do this by comparing the maximum physical batch sizes and throughputs between the non-private training and Opacus per-example clipping, called O-flat, which is the most used DP-SGD implementation. Additionally, we identify the reasons for the higher computational cost of DP-SGD through profiling.

### 4.1. Maximum Physical Batch Size

We first compare how many samples we can process in one physical batch before running out of memory. In Figure 2, we can see that the model size matters as its parameters must be stored in memory, and their gradients must also be stored for accumulation. As there are millions of parameters, it becomes increasingly expensive to store them in memory. As expected, due to its implementation of calculating the

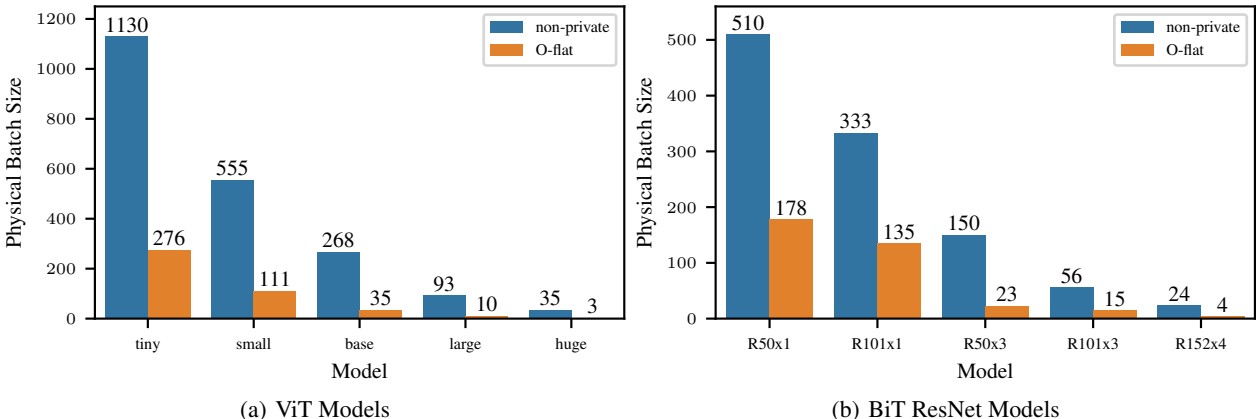

*Figure 2.* Maximum achievable physical batch size by the different model sizes on A100 GPU (40 GB) before they reach Out Of Memory (OOM) Error. The model sizes grow from left to right. To check the number of parameters of each size, refer to Table 1.

gradients for each sample, the per-example clipping fits fewer instances.

The difference in throughput between non-private and private training grows with respect to the model size. In Figure 2(a), the difference for the huge model size is already 11x times the number of fitted samples, and the private model can only fit three samples, getting closer to just calculating one sample at a time and accumulating that for a large logical batch, which is very inefficient. This is problematic as a bigger model would be unable to fit even one sample.

From the experiments we also verified that the use of physical batches does not affect the model performance; each model size will reach a different accuracy than the others, but the same model size will always reach the same performance independent of the physical batch size.

### 4.2. Throughput Comparision

The batch size alone does not tell the whole story, as the main interest is how much slower the private training is. Figure 3 shows the relative difference between mean throughput, which is how faster the non-private training is and how much more expensive the private one is. It is consistent with Figure 2 in that as the model size grows, the difference between how many samples can be fitted and the throughput difference will grow larger.

**ViT** The throughput difference between Opacus and a non-private baseline for the ViT model (see Figure 3(a)) does not spike and grows steadily, which is interesting considering how big the relative difference is in physical batch size. Using the previous example with the ViT huge model, the throughput difference is about ×3.1, while we can fit 11 times more samples in the non-private implementation.

**ResNets** On the other hand, in Figure 3(b), the ResNet mod-

els have different behavior. They do have spikes of growth as the model size grows. The contrast in Figure 3 between ViT and BiT ResNet models is due to the architecture and types of layers. The parameter space grows as the width factor (see Table 1) for the ResNets, so the ×3 makes the neural network wider by a factor of three. Based on our results, the width of the layers affects throughput much more than the depth of the network. They have comparable throughput with the same width and different depths, but increasing the width will make the model in the private setting much slower and reduce the maximum batch size significantly.

**How much does finding the maximum physical batch size matter?** In Figure 4, we display the relative throughput as a percentage by dividing the throughput at a particular physical batch size by the maximum achievable throughput. We see that as the physical batch size increases, the throughput will grow as expected, but at some point there is no signifant further improvement in throughput from using a larger physical batch size.

Practitioners may estimate the optimal batch size based on available memory and performance trade-offs. It is not crutial to set the physical batch size to the maximum possible but a good enough value be fine. Typically, smaller batches are limited by data loading speeds, but as batch size increases, computation becomes the limiting factor.

### 4.3. Reasons for the Increase in Computational Cost

Giving a detailed breakdown of low-level operations associated with DP is challenging. However, using GPU profiling tool NVIDIA Nsight System we can identify three aspects which constitute to the majority DP overheads. Firstly, due to its larger memory footprint, DP-SGD is able to consume smaller physical batches than its non-private counterpart. This results in larger amount of smaller low-level kernel

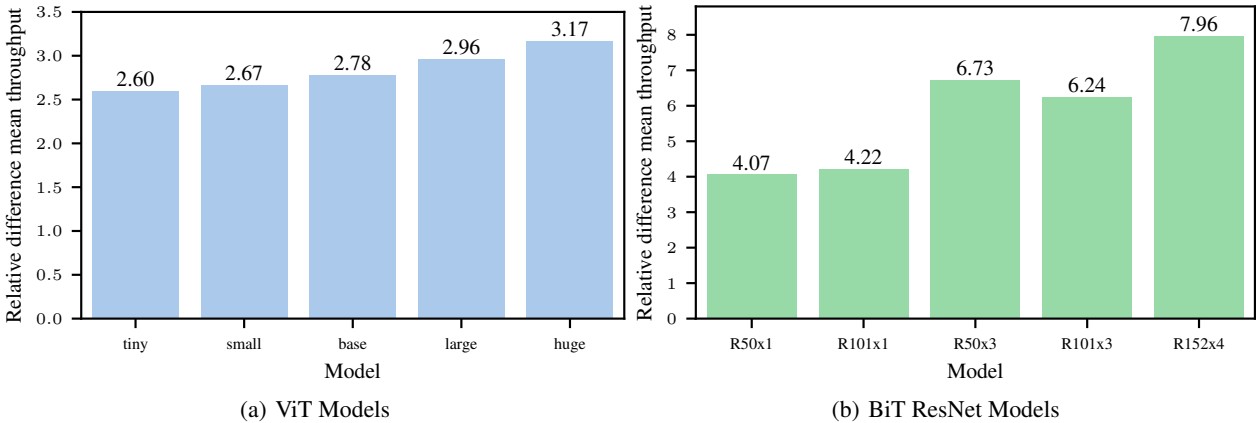

(a) ViT Models

(b) BiT ResNet Models

*Figure 3.* Relative difference between mean throughputs between Opacus per-example clipping and the non-private baseline. It is defined as private-throughput/non-private-throughput. It shows how many times private training is more expensive. These experiments are executed on one A100 GPU.

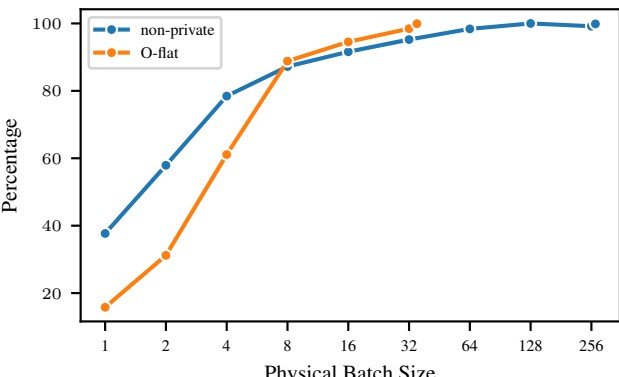

*Figure 4.* The relative difference with the throughput at the maximum batch size for the ViT base model on A100.

*Table 2.* Maximum physical batch size reachable for each clipping method, for the two GPU architectures we are comparing, for the ViT base model.

| CLIPPING MODE | V100 | A100 |
|---|---|---|
| NON PRIVATE BASELINE | 216 | 268 |
| O-FLAT (OPACUS) | 28 | 35 |
| GHOST (PRIVATE VISION) | 203 | 257 |
| MIX GHOST (PRIVATE VISION) | 203 | 257 |
| BK GHOST (FASTDP) | 189 | 209 |
| BK MIX GHOST (FASTDP) | 189 | 209 |
| BK MIX OPT (FASTDP) | 189 | 209 |

### 5.1. Optimized algorithms

First, we evaluate the more efficient implementations that have been described in Section 2.2 using the ViT base model. We chose it as our benchmark model because the middle model size is large enough to test our hypothesis but less extensive than it would take too much time to train. The implementations do not support the BiT ResNet due to their custom weight standardization layer.

**Maximum physical batch size** Table 2 compares the maximum physical batch size for both available GPUs. The maximum physical batch size is larger for the optimizations of DP-SGD than for Opacus because they do not require per-example gradients. Thus, the optimizations allow training much larger models without running out of memory. The maximum physical batch size using Private Vision library is the one that comes closest to the non-private baseline. In general, we can see that the methods are consistent within implementations, with Private Vision and the FastDP reaching the same maximum physical batch size no matter the clipping mode. As expected, the A100 achieves consistently

calls which leads to slightly lower utilisation of the GPU compute. At very small batch sizes even the kernel launch overheads can become a notable factor for slowdown. Secondly, the computation of per-example-gradients introduces significant overhead in the backward pass as it cannot be parallelised as in batched gradient computation. This is the most prominent cause of the total overhead. Finally, an additional DP-optimizer step that clips and accumulates the per example gradients, which is not present in the non-dp algorithm, needs to taken after each physical batch (see Table A3).

## 5. Decreasing the computational cost

This section analyzes the different strategies for training with DP-SGD more efficiently. We evaluate both algorithmic and hardware optimizations and their combinations.

higher maximum physical batch sizes than the V100 due to the larger amount of VRAM.

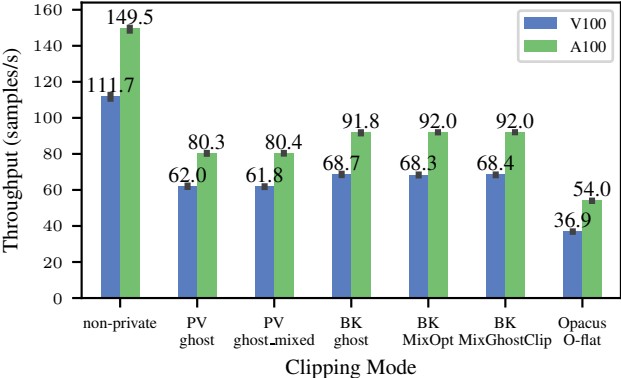

*Figure 5.* Throughput using the maximum batch size for each clipping algorithm. It compares the executions for both V100 and A100, for the ViT Base model.

**Throughput Comparision** Figure 5 displays the throughput for each clipping algorithm for each available GPU. To understand it better, we also compute the relative difference between the mean throughput. By using better hardware components, in this case, improving from a V100 to an A100 GPU has an increment of ×1.3 times in throughput for all clipping methods. The one that benefits the most is the per-example clipping by Opacus with a ×1.46 improvement in throughput. It is because of Opacus-specific optimizations. Their implementation is optimized to vectorize the virtual batches and get the most out of the processing unit (Yousefpour et al., 2021) to compensate for the per-example clipping. Also, we base our virtual batching module on Opacus, being specially optimized for the Opacus implementation, yielding better results with better hardware. The other implementations will have benefits similar to those of non-private training. The clipping methods in both GPU architectures will have the same relative throughput regarding the non-private performance. Private Vision gets closer to the non-private baseline physical batch size, but Book Keeping is closer to its throughput with a smaller physical batch size (see Figure 6).

Without sacrificing utility, these optimizations offer an alternative to the original per-example clipping algorithm. Even though Book Keeping performs better, it is by a very narrow margin. Consequently, Private Vision and FastDP remain viable options for implementing ghost clipping. The difference between the two algorithms is the second backward pass over the neural network. Since the Book Keeping trick avoids doing the second backward pass through the network, it has a higher throughput at a small memory cost.

Mixing ghost clipping does not yield any improvement be-

cause it determines whether it should apply ghost or per-example clipping, which depends on the size of the inputs and the parameter space. If the dimensions are large enough, the ghost technique will be more expensive (Bu et al., 2022). In a ViT model, the dimensions change less than in a convolutional network. Therefore, despite continually evaluating which method to apply, it always uses ghost clipping. However, if applied to a ResNet model, it should outperform ghost clipping, as it is optimized for convolutional layers. It could not be tested on BiT ResNet models used in this study due to incompatibilities with the Private Vision and FastDP, preventing an assessment of mixed optimization methods.

## 5.2. JAX framework

In this section we compare the performance of JAX with all other DP-SGD frameworks (all of them are based on PyTorch). To make a fair comparison between frameworks, we implemented Poisson subsampling and virtual batching, based on the Opacus implementation.

**Compilation time** Comparing JAX to PyTorch requires taking the compilation time into account that the DP-SGD implementations in PyTorch do not utilize. There is no straightforward way of calculating the compilation time, but we measure it as the duration to process the first physical batch. The execution times for each batch shows that the first one takes much more time than the others, therefore including the compilation time (Figure A.1). To provide a fair comparison, we also implemented a non-private JAX training using the same virtual batching as PyTorch.

**Throughput comparision** In Figure 6, both JAX implementations start at the same point, since by using virtual batching, they are processing just one sample at a time and getting the gradients from the compiled function. The non-private benefits more from a larger batch size, although its throughput decreases at 128 instances. The compilation time (see Figure A.1) grows with the batch size. For the private model, it takes more time since the compiled function is more complex than the non-private counterpart. It includes expanding the dimensions and clipping the gradients, while the non-private directly computes the gradient of the whole mini-batch.

JAX per-example clipping reaches a higher physical batch size, and the throughput is always higher than its Opacus counterpart. However, the Book Keeping Ghost Clipping implementation is closer to the JAX private version. They are the same until the physical batch size of 32, but even then, the variance in the JAX execution is high, meaning that it can even be below the BK-ghost throughput. JAX private implementation throughput drops when the batch size doubles to 64. This comparison would be a reason to use Book Keeping ghost clipping for Vision Transformer models over JAX implementation. Its throughput is com-

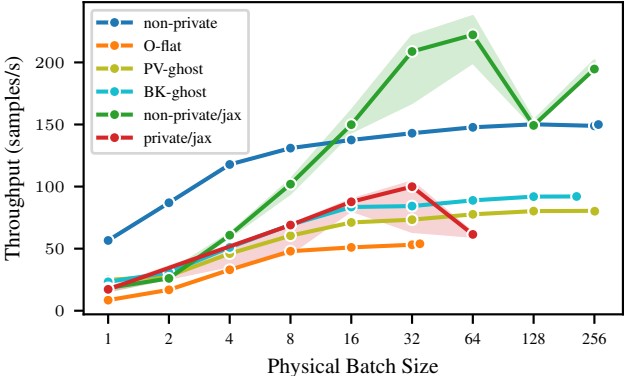

*Figure 6.* Comparison of the throughput as a function of the physical batch size between the JAX and PyTorch clipping algorithms on A100 GPU. Only the ghost implementations from Private Vision and Book Keeping are used, not the Mix algorithms, since they have the same performance. The estimator is the median, and the error bars are the 95% confidence interval using bootstrapping. All methods use the same estimator and confidence interval, and JAX is the only one in which the throughput spreads significantly across runs.

parable to the execution of compiled code from JAX while reaching a larger batch size.

JAX private model starts with a similar throughput as the non-private model and then steadily grows until it surpasses the Torch throughput after a physical batch size of 16. After that, JAX will always have better throughput, except at a physical batch size of 128, where JAX throughput drops and is the same as the PyTorch non-private. We ran the experiment multiple times, and at the 128 size, it has a low variance, with always the same throughput value. Another difference between the two frameworks is the variability in the experiments. PyTorch runs are remarkably consistent in having a low variance, and the same throughput result is expected every time for a fixed seed.

JAX executions are more variable than those of PyTorch, likely due to its sensitivity to HPC environment fluctuations and accelerator stochasticity, as noted in Figure A.1. Another contributing factor is JAX's asynchronous dispatch method, which complicates time benchmarking by issuing a promise rather than immediate results, concealing Python overheads.

**Poisson sampling** Using JAX for DP introduces complexities, particularly around subsampling which is crucial for privacy accounting. Implementing Poisson subsampling results in variable batch sizes in JAX; changes in batch size require JIT to recompile, leading to graph retracing which is costly and contributes to execution run variability, as discussed by Chua et al. (2024).

**Comparison with PyTorch** Although compiling PyTorch is possible, we could not see any improvements in terms of speed-up. While compiling the non-private model worked, the speed-up gained was minimal and, in the end, even lower if we consider the compilation time. PyTorch also recompiles after a batch size change. While trying to compile, PyTorch falls back to predefined CUDA operations that are already optimized. In the case of the private setting, the compilation does not recognize Opacus hooks and continues the execution without compiling them.

Leveraging the same kernels to support the private hooks and avoid the compilation would require massive engineering work of writing special kernels for each specific private case. On the other hand, JAX will compile the JIT functions in XLA, but it does not fall back to the kernels, making it more generalizable (Subramani et al., 2021).

JAX reaches a higher accuracy in Table A4 than its PyTorch counterpart. We are still evaluating what is causing the significant change, even when using the same hyperparameters and training for just two epochs. The difference is more prominent for the non-private case, but we still see the DP utility cost.

### 5.3. Lower precision

We consider using lower precision to speed up computational times. We evaluate the use of Tensor Core 32 (TF32) for training. It has 10 bits for precision, with eight range bits, the same as 32 single precision (FP32)(Kharya, 2020). Using lower precision can have benefits exactly where DP training struggles: it requires less memory, uses fewer bits to represent the data, and its operations are optimized for GPU, making them much faster (NVIDIA, 2023). It is specially optimized for the A100 GPU and unavailable for the V100, so the comparison will be between training on the A100 with and without TF32.

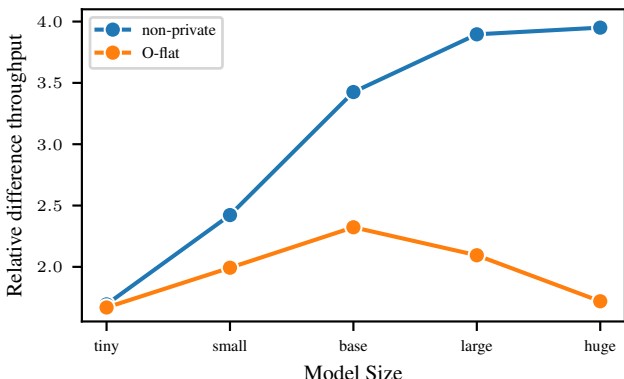

*Figure 7.* Relative difference in mean throughput between TF32 and FP32 Training for ViT Models.

**Experimental results** In Figure 7, the relative difference between mean throughput between runs with TF32 and FP32. For non-private training, throughput increases with model size, peaking at nearly four times the speed for the ViT Huge model. Then, the private training throughput increases for the smaller models, but it goes down again as the model size grows after the base size. The ViT Base model presents the optimal speed-up, suggesting that the best throughput can be obtained using a middle-size model. One that is too small does not gain much from TF32, and the larger ones are too expensive. If we compare the non-private, the growth between the large and huge model sizes is minimal. The model size is the bottleneck, and the private one is the most affected by it due to the nature of the clipping algorithm. However, this remains open for further investigation on why the private training goes up and down with the model size in lower precision. Regarding the memory advantages by TF32, we could not see an improvement. Both models, with and without TF32, could fit the same number of instances.

**Concerns regarding TF-32 under DP** There are two concerns with using lower precision in DP deep learning: its effects on utility and privacy. For the first issue, using lower precision may affect utility, as it is less precise. We did not find a significant decay in the accuracy of the models; it differs by decimal points at the $\times 10^{-6}$ precision (See Table A4 in the Appendix). For example, private training using the optimal number of training steps, both models reach the same accuracy of $0.879$ with a significant $\sim 2.3$ speed-up thanks to lower precision.

The second concern about privacy was first highlighted by Mironov (2012), but no fully satisfactory solution exists yet. The problem is that theoretical guarantees for most algorithms assume computations are performed on reals, and finite precision arithmetic can lead to actual violations. Discrete mechanisms (e.g. Canonne et al., 2020; Agarwal et al., 2021) that avoid the theoretical challenges exist, but they are often less convenient and may lead to loss of utility, especially in low precision settings. The efficiency of different discrete mechanisms in TF-32 is an interesting topic of further research.

## 6. Multi-GPU Training

This subsection will look at another angle to train deep learning with DP faster: increasing the computational resources enough to decrease the training time. This is relevant when training cost or resource constraints are less important than the time to train a new model.

### 6.1. Experimental Setup

We utilize V100 GPUs on HPC nodes that have 4 GPUs per node. The other experimental setting is identical to the one

in Section 4. Results for utilizing up to 24 A100 GPUs can be found in the Appendix Figure A.3.

We focus on comparing the scaling behaivour between the non-private baseline that uses PyTorch and the DP-SGD implementation using Opacus. Both frameworks provide mature tooling for distributed training. Although the other clipping methods do not break, we are still determining if they are correctly handling the distributed gradient computation and clipping or if distributed training is even supported.

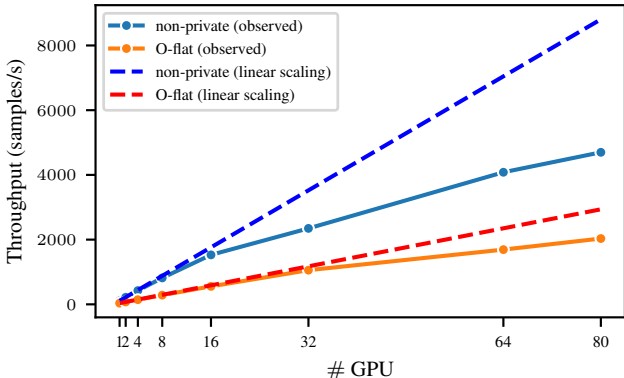

*Figure 8.* Comparison between the throughput by scaling the number of GPUs with more nodes for the non-private and Opacus training with the ViT base model on V100 GPUs. The dashed line is the ideal growth if it were linear.

### 6.2. Experimental Results

Figure 8 shows the throughput increase as we add more computational resources, going from one to 80 GPUs. The plot shows that the throughput does not grow linearly and starts changing from the ideal after using more than one node (i.e. when using more than 4 GPUs). The communication inside the node is fast, but the communication between nodes will always be slower. The bottleneck is the bandwidth, and it prevents the model from scaling linearly. Notably, it affects the non-private training baseline much more, while the private scales better. For the 80 GPUs, the private training achieves 69.2% of the ideal throughput, and the non-private training only achieves 53.3%. Private training scales better because it is slower and only sometimes saturates the network with updates.

If we use Amdalh's law to compare the parallelism percentage for each case, we can see that in the private case, we achieve a 99.5% parallelism compared to a 98.9% in the non-private case (See Figure A.4 in the Appendix).

# 7. Conclusion

While DP-SGD is significantly more costly than non-private training, we identified feasible speed-ups that are often easy to apply but have some drawbacks. These are: (i) More efficient implementations of DP-SGD which additionally decrease the memory footprint (allowing for training larger models). However, the implementations are not as mature as Opacus and do not support all types of neural network layers (yet). (ii) JAX which processes samples faster than PyTorch but looses the advantage through frequent re-compilations when utilizing proper Poisson sampling, does not offer a comprehensive DP-SGD implementation as PyTorch and runs not as stable as PyTorch. (iii) Lower Precision using TF-32 which increases the throughput but the implications on the theoretical guarantees of DP-SGD need to be explored in future work. Finally, we found that distributed computing using DP-SGD scales better than non-private training and allows for fast training of models.

# Acknowledgements

This work was supported by the Finnish Ministry of Education and Culture and CSC - IT Centre for Science (Decision diary number OKM/10/524/2022), the Research Council of Finland (Flagship programme: Finnish Center for Artificial Intelligence, FCAI, Grant 356499 and Grant 359111), the Strategic Research Council at the Research Council of Finland (Grant 358247) as well as the European Union (Project 101070617). Niki Loppi contributed under the NVIDIA AI Technology Center (NVAITC) Finland program. Views and opinions expressed are however those of the author(s) only and do not necessarily reflect those of the European Union or the European Commission. Neither the European Union nor the granting authority can be held responsible for them. This work has been performed using resources provided by the CSC – IT Center for Science, Finland. We thank Joonas Jälkö for helpful discussions regarding implementing DP-SGD with JAX.

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

# A. Training Details

## A.1. Framework and implementation compatibility

Table A1 shows the corresponding support between implementations and clipping methods. We also include general frameworks, like PyTorch and JAX. Opacus, PrivateVision, and FastDP are PyTorch based implementations.

*Table A1.* Clipping optimization and the library or framework that implements it.

| CLIPPING MODE | PYTORCH | OPACUS | PRIVATEVISION | FASTDP | JAX |
|---|---|---|---|---|---|
| NON-PRIVATE | ✓ | | | | ✓ |
| PER-EXAMPLE | | ✓ | | | ✓ |
| GHOST CLIPPING | | | ✓ | ✓ | |
| MIX GHOST | | | ✓ | ✓ | |
| MIX OPT | | | | ✓ | |

## A.2. Models

- Vision Transformer (ViT) (Dosovitskiy et al., 2021). Taken from `https://huggingface.co/timm/vit_base_patch16_224.augreg2_in21k_ft_in1k`

- Big Transfer BiT-ResNet (Kolesnikov et al., 2020). Taken from `https://github.com/google-research/big_transfer`

## A.3. Hyperparameters

We use the hyperparameters obtained on request from Tobaben et al. (2023). The hyperparameters for both models are in Table A2. We do not optimize them futher, as model utility is not the main objective in this work.

*Table A2.* Hyperparameters used for each model architecture.

| MODEL | TRAINABLE PARAMETERS | EPSILON | DELTA | LEARNING RATE | MAX GRAD NORM |
|---|---|---|---|---|---|
| VIT | ALL | 8 | $2.04e^{-5}$ | 0.0003 | 4.63 |
| BIT-RESNET | ALL | 8 | $2.04e^{-5}$ | 0.00098 | 6.53 |

# B. Additional Results

This section provides additional figures that supplement the findings in the main text.

*Table A3.* Average processing time for each section of the algorithm. We are comparing the non-private and Opacus per-example clipping on A100, with the same physical batch size. It is calculated with NVIDIA Nsight Systems. All the measurements include the syncronization time, which is needed for the profiling, but adds additional time that is not part of the normal execution. All values are in milliseconds.

| SECTION | PYTORCH NON-PRIVATE | OPACUS PER-EXAMPLE |
|---|---|---|
| FORWARD | 81.14 | 101.53 |
| BACKWARD | 163.85 | 681.48 |
| CLIP AND ACCUMULATION | 0 | 26.76 |
| OPTIMIZER STEP | 38.17 | 99.65 |

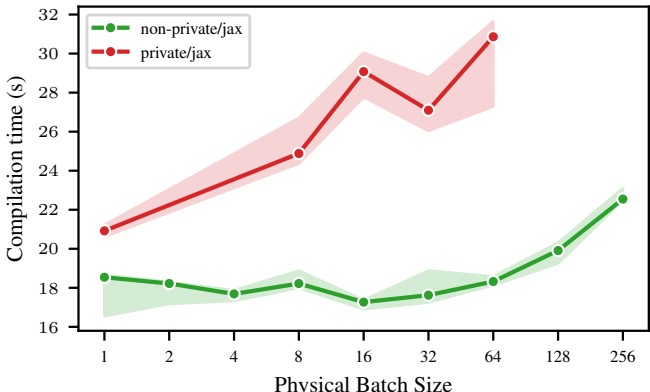

*Figure A.1.* Compilation time in seconds as a function of the physical batch size for JAX experiments for the ViT Base model on A100. The estimator is the median and the error bars are the $95\%$ confidence interval using bootstrapping.

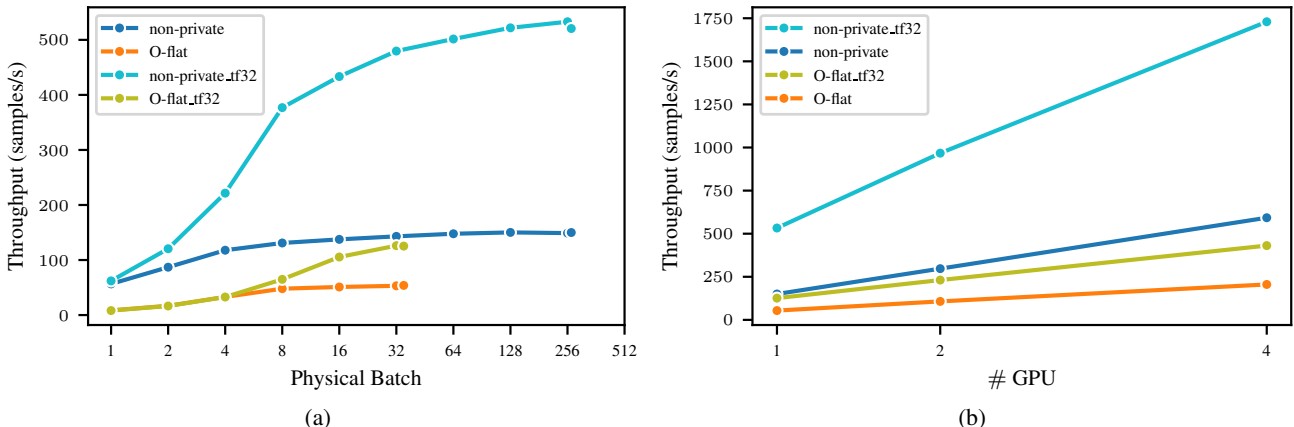

(a)                                                                                      (b)

*Figure A.2.* Combining distributed training with the use of lower precision TF32 for the ViT base model on A100. (a) Throughput for one GPU; (b) Throughput for multiple GPUs.

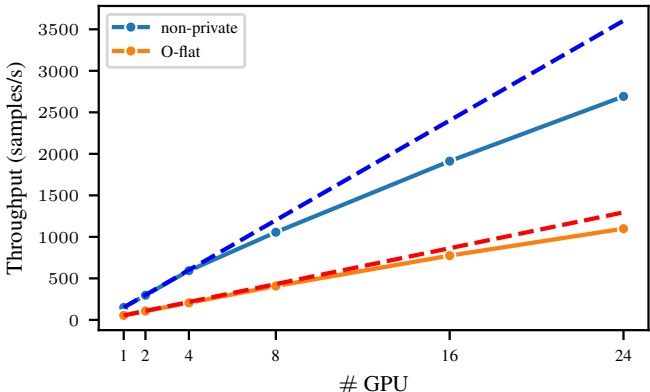

*Figure A.3.* Comparison between the throughput by scaling the number of GPUs with more nodes for the non-private and Opacus training with the ViT base model on A100 GPUs. The dashed line is the ideal growth if it were linear.

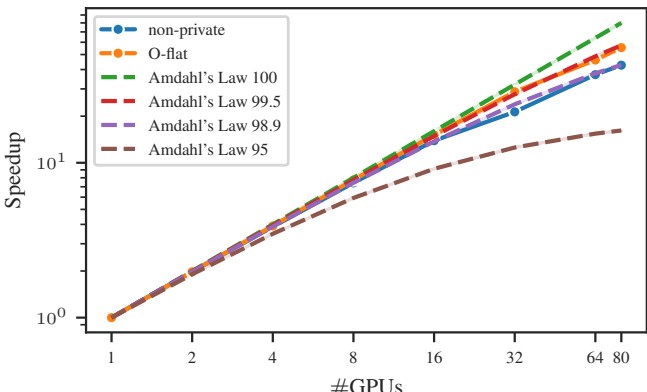

*Figure A.4.* Comparison between the throughput in our experiments and the theoretical Amdahl's Law. Both axis are in log scale. In the distributed setting, private training achieves a 99.5 % of parallel processing, with a 50 times speed up than single processing.

*Table A4.* Mean accuracy for CIFAR-100 test set for each clipping mode for the ViT models on A100 after training for two epochs. All use the ViT hyperparameters from Table A2. While this work does not focus on the model's utility, having their results still allows us to compare them. Using optimal hyperparameters for DP causes low utility in non-private training. The use of TF32 as a lower precision mode does not affect the model's utility. We are still analyzing why JAX accuracy is significantly higher in both cases, even when it uses sub-optimal hyperparameters.

| CLIPPING MODE | TEST ACCURACY |
|---|---|
| NON-PRIVATE | 0.2285 |
| NON-PRIVATE/TF32 | 0.2301 |
| NON-PRIVATE JAX | 0.8273 |
| O-FLAT OPACUS | 0.7096 |
| O-FLAT OPACUS/TF32 | 0.7094 |
| O-FLAT JAX | 0.8009 |
| PV-GHOST | 0.6978 |
| BK-GHOST | 0.7341 |

