# OpenReview forum: "Towards Efficient and Scalable Training of Differentially Private Deep Learning"
_ICML.cc/2024/Workshop/WANT — WANT@ICML 2024 Poster_

### Official Review · Reviewer_5P2H · 2024-06-10
**The paper analyzes different implementations, which helps optimize method performance. Despite the lack of various metrics and further improvements, the efficiency of DP-SGD is worth studying.**

**Confidence:** 3

**Summary:**

This paper studies the efficiency problem of DP-SGD and locates the critical factors that lead to high computational costs. With these analyses, the paper concludes the directions to improve DP-SGD for efficiency. The paper provides experimental contributions in various levels of code implementation.

**Strengths:**

1.	There are sufficient experimental results to support the claims.
2.	The advantages and disadvantages of various implementations are analyzed. Some critical points are located, which helps the community apply or improve them.

**Weaknesses:**

1.Lack of model performance reports (e.g., accuracy) under different implementation conditions. Different implementations could not always show similar results.

---

### Official Review · Reviewer_yvJ4 · 2024-06-13

**Confidence:** 3

**Summary:**

This paper presents a set of comprehensive experiments and analysis exploring multiple aspects of scaling and efficiency of training with DP-SGD. The authors benchmark multiple frameworks and implementations, for various model sizes. The experiments are conducted on a vision dataset (CIFAR-100).

**Strengths:**

1. The experiments are comprehensive, exploring multiple aspects of efficiency while training with DP-SGD.
2. I believe that the insights will be very useful to practitioners.

**Weaknesses:**

1. While I understand the focus of the study is not finding the best utility, end task performance is not reported for any experiment, which makes interpretations of some results hard to contextualize.

**Suggestions:**

1. There is no mention of availability of code - the poisson subsampling implementation for other frameworks will be useful to many.
2. As mentioned before, I would appreciate some numbers showing final performance, including on some other modalities like text.

---

### Official Review · Reviewer_aKwa · 2024-06-13
**Serious empirical study on improving DP-SGD throughput**

**Confidence:** 3

**Summary:**

The authors propose an empirical study of the performance bottlenecks of differentially private stochastic gradient descent (DP-SGD). Besides the per-sample clipping, the authors identify by profiling that per-sample gradients introduce significant overhead in training compared to non-private training. They benchmark different strategies to reduce this overhead including ghost clipping, book keeping, implementation in JAX, and lower precision. Experiments show that these techniques increase throughput and scale with several GPUs.

**Strengths:**

- The paper is clearly written and well organized;
- The work covers different methods to improve DP-SGD throughput. It provides empirical evidence of their respective success.

**Weaknesses:**

The paper has no obvious weakness. The empirical study seems well conducted. The novelty and the interest of its findings may be emphasized more.

The paper raises the following question:
- It will be interesting to provide more details, even assumptions, on why the JAX implementation is outperforming the one with Pytorch;
- I will be curious to know if besides exploring JAX implementation, there are other avenues for accelerating DP-SGD, from new algorithms to other implementation improvement, possibly relying on tailored CUDA kernels.

---

### Meta-Review · Area_Chair_V76z · 2024-06-16

**Recommendation:** Accept (Poster)
**Confidence:** 4

**Metareview:**

## Strengths
* This paper is well written
* The paper contains several useful insights for the community about how to improve DP-SGD performance
* Several comprehensive experiments support the claims, and are analyzed in detail
## Weaknesses
* The paper lacks results about end task performance, that could help provide insights about the
  tradeoff between computing cost and task performance

The general sentiment appears to be rather positive, I recomment acceptance as a poster presentation.

---

### Decision · Program_Chairs · 2024-06-17

**Decision:**

Accept (Poster)

**Comment:**

We thank the authors for their time and contribution to WANT and we are pleased to share that after the reviewing process the paper has been accepted. Congratulations! We encourage the authors to consider reviewers' feedback for the improvement of the camera-ready version. We hope to see you in person at the workshop and brainstorm on efficient training research together!